# Strain-Dependent Variability in Ochratoxin A Production by *Aspergillus* spp. Under Different In Vitro Cultivation Conditions

**DOI:** 10.3390/microorganisms13122850

**Published:** 2025-12-15

**Authors:** Zuzana Barboráková, Dana Tančinová, Juraj Medo, Silvia Jakabová, Georg Häubl, Günther Jaunecker, Zuzana Mašková, Roman Labuda

**Affiliations:** 1Institute of Biotechnology, Faculty of Biotechnology and Food Sciences, Slovak University of Agriculture in Nitra, Tr. A. Hlinku 2, 94976 Nitra, Slovakia; zuzana.barborakova@uniag.sk (Z.B.); juraj.medo@uniag.sk (J.M.);; 2Institute of Food Sciences, Faculty of Biotechnology and Food Sciences, Slovak University of Agriculture in Nitra, Tr. A. Hlinku 2, 94976 Nitra, Slovakia; silvia.jakabova@uniag.sk; 3Romer Labs Division Holding GmbH, Technopark 5, 3430 Tulln an der Donau, Austria; georg.haeubl@dsm-firmenich.com (G.H.);; 4MycoLAB RL GmbH, Technopark 1C, 9430 Tulln an der Donau, Austria; roman.labuda@mycolab.at

**Keywords:** *Aspergillus*, ochratoxin A, HPLC, cultivation media, cultivation conditions

## Abstract

The aim of this study was to investigate differences in the dynamics of ochratoxin A (OTA) production by various *Aspergillus* isolates under different cultivation conditions. Nine strains representing *A. westerdijkiae*, *A. ochraceus*, *A. sulphureus*, *A. carbonarius*, and *A. albertensis* were tested on malt extract agar (MEA), Czapek yeast extract agar (CYA), potato dextrose agar (PDA), and yeast extract sucrose agar (YES). Cultivations were performed at 18 °C, 22 °C, 25 °C, and 30 °C, and OTA production was monitored on the 6th, 10th, 14th, 21st, and 30th days using HPLC analysis. OTA yields strongly depended on the producing strain, with significant variability even among isolates of the same species. The most productive strain was *A. ochraceus* from cereals with a maximum concentration of 848 µg g^−1^ OTA, followed by two isolates of *A. westerdijkiae* from grapes of Slovak origin (591 and 479 µg g^−1^), and *A. sulphureus* from soil (546 µg g^−1^). In contrast, *A. carbonarius* strains showed the weakest OTA production. Across media, YES supported the highest toxin levels, whereas the most favourable cultivation temperatures were 18 °C and 25 °C. Each strain reached its production maximum at different time points, highlighting the strain-specific nature of OTA biosynthesis.

## 1. Introduction

Mycotoxin contamination of food and feed represents a high risk for human and animal health [1] and is considered one of the most dangerous contaminants of food and feed [2].

Ochratoxin A (OTA) is one of the most important mycotoxins. OTA was first described by Van der Merwe et al. in 1965 (in [3]) as a toxic metabolite of the fungus *Aspergillus ochraceus* in a corn meal in South Africa [4]. Since it was discovered, it has been ubiquitous as a natural contaminant of moldy food and feed [5].

Biosynthetically, it is a pentaketide derived from the dihydrocoumarin family coupled with β-phenylalanine [6]. OTA is an immunosuppressant fungal compound, produced by a toxigenic species of *Aspergillus* and *Penicillium* fungi in a wide variety of climates and geographical regions [6,7]. OTA is produced in foodstuffs by *Aspergillus* section *Circumdati* (e.g., *A. ochraceus*, *A. westerdijkiae*, *A. steynii*), *A.* section *Nigri* (e.g., *A. carbonarius*, *A. foetidus*, *A. niger*, *A. lacticoffeatus*, *A. schlerotioniger*, *A. tubingensis*), and *A.* section *Flavi* (e.g., *A. albertensis*) [8], mostly in subtropical and tropical areas. It can also be produced by *Penicillium verrucosum* and *P. nordicum*, notably in temperate and colder zones [9]. *P. verrucosum* and *A. ochraceus* are able to synthesize OTA at lower temperatures than *A. niger* and *A. carbonarius* [10].

Higher amounts of OTA were produced on wheat than on other substrates, including maize, peanuts, rapeseeds, and soybeans [11]. OTA is frequently found in foodstuffs of both plant origin (e.g., cereal products, coffee, vegetables, licorice, raisins, wine, beer, wine and grape juice, dried vine fruits, nuts, cacao products, spices) and animal origin (e.g., pork/poultry) [9,12].

Dietary exposure to OTA represents a serious health issue and has been associated with several human and animal diseases, including poultry ochratoxicosis, porcine nephropathy, human endemic nephropathies, and urinary tract tumors in humans [3]. Many experimental studies have shown that the ingestion of OTA can have several consequences, such as nephrotoxic, hepatotoxic, neurotoxic, teratogenic, and immunotoxic effects [13]. The International Agency for Research on Cancer (IARC) classified OTA as a possible human carcinogen—Group 2B [14].

The main factor affecting fungal growth and OTA production is water activity (a_w_) followed by temperature, although the effect of substrate should not be disregarded. All these factors are continuously interacting in the environment; therefore, fungal ability to grow and produce mycotoxins is affected by a complex combination of parameters. Moreover, optimum conditions for fungal growth are usually different from those for mycotoxin production [15]. The influence of these factors has been studied by culturing fungi under different environmental conditions in suitable growing substrates, or native or sterilized foodstuffs, with a significant impact on OTA-producing fungi growth and mycotoxin production [10]. Despite the effects of environmental conditions being well studied, variability between strains within the same species group has received less attention [16,17]. In particular, the dynamics of OTA production over time and its dependence on specific cultural conditions remain poorly understood, limiting effective strategies to mitigate food contamination. These gaps prevent accurate assessment of the risks associated with OTA in commodities such as grapes and cereals.

Recent studies [16,17] suggest that individual isolates may differ markedly in their ochratoxigenic potential, even when belonging to the same species. However, comprehensive comparisons of multiple *Aspergillus* isolates cultivated under strictly identical conditions remain limited. Furthermore, time-resolved analyses following OTA formation over extended incubation periods are scarce, although available evidence indicates that OTA accumulation, degradation, or metabolic downregulation may vary significantly over time. Integrating both maximum OTA concentrations and cumulative toxin production across the cultivation period may therefore provide a more complete understanding of OTA biosynthesis dynamics under controlled conditions.

Importantly, although many studies have examined the influence of temperature, a_w_, substrate composition, or fungal species on OTA production, comparative analyses of multiple isolates cultivated under fully standardized conditions remain scarce. Moreover, strain-level differences in long-term OTA production dynamics (e.g., cumulative toxin output) are not well characterized, despite evidence that such differences may substantially affect contamination risk. This gap highlights the need to systematically evaluate strain-dependent variability in OTA biosynthesis under controlled conditions.

The aim of this study was to quantify the variability of OTA production by nine *Aspergillus* strains (sections *Circumdati*, *Nigri*, and *Flavi*) on four culture media (YES, CYA, PDA, and MEA) at different temperatures (18, 22, 25, and 30 °C) and to monitor its dynamics over 30 days in order to identify high-risk strains and optimal conditions for OTA biosynthesis. This approach provides comprehensive data to improve risk assessment strategies and prevent OTA contamination in the food chain. It should also be noted that the present study was performed exclusively under controlled laboratory conditions using pure fungal cultures. While this strictly in vitro approach allows precise comparison of strains and environmental parameters, it does not simulate biological interactions occurring in natural substrates, such as competition with other microorganisms, substrate-specific stressors, or ecological factors that may influence OTA production.

## 2. Materials and Methods

### 2.1. Strains of the Genus Aspergillus

Nine strains of the genus *Aspergillus* were used in this study (Table 1).

The isolates were identified based on macroscopic and micromorphological characteristics following the taxonomic frameworks of Samson et al. [18], Frisvad et al. [19], Pitt and Hocking [20], and Visagie et al. [21] (Table 2). Although molecular identification provides higher resolution within certain *Aspergillus* complexes, it was not performed in this study, as the primary aim was to compare OTA production dynamics among food-associated isolates rather than to resolve fine-scale taxonomy. Furthermore, additional molecular analyses cannot be conducted retrospectively, as several of the original isolates are no longer available in our laboratory collection. This limitation is further addressed in the Discussion.

### 2.2. Cultivation Media and Conditions

The strains of the genus *Aspergillus* were cultivated on Czapek yeast extract agar (CYA; [22]), malt extract agar (MEA; Himedia, India), potato dextrose agar (PDA; Himedia, India), and on yeast extract sucrose agar (YES; [23]). Petri dishes (plates) with a 90 mm diameter were used for cultivation. The plates were inoculated at the center with 10 µL of spore suspension prepared from *Aspergillus* strains grown on MEA at 25 °C for 7 days, in darkness. Spores were collected by rinsing the colony with physiological saline solution supplemented with Tween 80 (0.5%). For each *Aspergillus* strain a conidial suspension was prepared with a concentration of 10^5^ spores/mL. The number of spores was determined using an EVETM automatic cell counter (NanoEnTek, Seoul, Republic of Korea). Inoculated media were cultivated at 18 °C, 22 °C, 25 °C, and 30 °C in the dark conditions. Production of OTA was observed on the 6th, 10th, 14th, 21st, and 30th day of cultivation.

### 2.3. Sample Preparation and HPLC Analysis

A total of three agar plugs (0.5 g), were taken off by the cork borer (9 mm in diameter) from central and subcentral area (representing zones 1 to 3 according to Levin et al. [24]) of a culture of *Aspergillus*. Plugs were added into microtubes (2.0 mL, Eppendorf, Hamburg, Germany) and extracted in 1000 µL of ethyl acetate solution with 0.1% acetic acid. Extracts were mixed for five minutes by using an IKA MS 3 digital vortex and filtered through a 13 mm syringe filter 0.2 μL PTFE (VWR International, Radnor, PA, USA) into microtubes. 100 µL of prepared extracts were added into vials (2 ml, Agilent, Santa Clara, CA, USA) and then dried. Samples were reconstituted before HPLC (high performance liquid chromatography) analysis by adding 500 µL of acetonitrile solution: 0.1% H_3_PO_4_ (20:80, *v*:*v*). The analysis was performed in three replications.

HPLC was realized on the Dionex Ultimate 3000 system with a DAD detector by using a column Luna C18 (II), 250 × 3 mm, 5 μm (Phenomenex, Aschaffenburg, Germany). The solvents used were 0.1% H_3_PO_4_ in water (A) and acetonitrile (B) with a flow rate of 500 μL.min^−1^: 0 min 45% B; 1.5 min 45–60% B; 11.5 min 65% B; 14.5 min 65–45% B; 14.6 min 45% B. The limit of detection (LOD) was 0.01 µg g^−1^. Limit of quantification (LOQ) was defined as three times the LOD (0.03 µg g^−1^). As a reference standard, used BiopureTM ochratoxin A solid standards with a purity of 97.6 ± 2.4% were used (Romer Labs, Getzersdorf, Austria).

### 2.4. Statistical Analysis

Statistical analysis was carried out in the R statistical suite version 4.4.3 [25]. The general linear model (GLM) was used to determine the effect of factors on OTA concentration in media. Average concentration of OTA was analysed in the model Strain*Media*Temperature*Day with full interactions. The Shapiro-Wilk test was used for data normality evaluation. As the residues did not meet these criteria, logarithmic transformation (log10(OTA + LOD/2)) was used. The maximum concentration of OTA in time for each Strain*Media*Temperature was analysed in the same way. Area under the time curve (AUTC) was used to evaluate total OTA load during 30 days as the strains showed more or less rapid degradation of OTA. The AUTC values were calculated using the trapezoidal method, which approximates the integral of the measured variable over time. For a series of OTA concentration measurements *y_i_* taken at time points *t_i_* (*i* = 0, 1, …, *n*), AUTC was computed using the following equation.


(1)
$$AUTC=\sum_{i=1}^{n}\frac{\left(y_{i-1}+y_{i}\right)}{2}\times (t_{i}-t_{i-1})$$


Tukey HSD test on a significance level of α = 0.05 was used for evaluation of the difference between average values.

## 3. Results

### 3.1. The Contribution of Factors Affecting OTA Productions

The OTA concentration significantly varied, as shown in Figure 1. GLM analysis (Appendix A) of actual OTA concentration showed the highest effect of strain (R^2^ = 0.438), followed by an effect of cultivation media (R^2^ = 0.159) while temperature and timepoint were of substantially lower importance (R^2^ = 0.002 and 0.009) despite their strong significance. All interactions (2, 3, and 4 factors) included in the model were highly significant, explaining a considerable amount of variability (e.g., Strain*temperature, R^2^ = 0.086, η^2^ = 0.739). Despite low R^2^ values of temperature and timepoint, their interactions with other factors explained most of the remaining variability. As visible from Figure 1, the OTA concentration changed over time non-simultaneously for all strains and culturing conditions; moreover, OTA concentration declined in many cases. This led us to a comparison of maximum OTA concentrations during the incubation period instead of average concentration. In addition to the maximum OTA concentrations, we assessed the overall OTA accumulation during cultivation using the AUTC. This approach provides a cumulative indicator of OTA production throughout the entire incubation period rather than focusing only on peak values.

### 3.2. The Effect of Strain on OTA Production

Based on the results presented in Table 3 and Table 4, the ability to produce OTA varied considerably among the tested strains of the *Aspergillus* genus. The strains with the highest maximum OTA concentrations were not those with the largest AUTC values. The highest AUTC value (9123) was achieved by the *A. westerdijkiae* (no. 9) with a maximum concentration of 591 µg g^−1^ on YES medium. *A. sulphureus* strain CBS 550.65 (no. 7), confirming its potential as a strong OTA producer with AUTC value 8348. Strain *A. ochraceus* (no. 2) also achieved the best peak concentration of 848 µg g^−1^; however, production started late, and the AUTC value was not the highest one. Other strains maintained OTA synthesis over a longer period of time, resulting in a higher OTA load. Among the *A. ochraceus* strains, isolate no. 2 (Biomin, Tulln an der Donau, Austria) was substantially more productive, reaching its maximum on the 30th day of cultivation, whereas isolate no. 1 (IFA, Vienna, Austria) produced only low OTA amounts. The *A. albertensis* strain (no. 3) produced moderate OTA amounts. In contrast, the *A. carbonarius* strains (nos. 4–6) were the weakest OTA producers.

The AUTC area and the average maximum OTA production showed a similar pattern in some isolates, whereas in others they differed. A clear difference was observed in the isolate *A. westerdijkiae* (no. 8). The highest average OTA concentration was recorded on day 10 of cultivation on YES medium at 18 °C (479.73 µg g^−1^), while the highest AUTC value was obtained on YES at 30 °C. This was due to an earlier onset of toxin production at 30 °C—by day 6 it was more than fourfold higher compared to 18 °C (38.12 µg g^−1^ vs. 169.66 µg g^−1^)—and the OTA concentration remained detectable even on day 30 at 30 °C (162.66 µg g^−1^), whereas at 18 °C it dropped below the detection limit.

### 3.3. The Effect of Media and Temperature on OTA Production

A detailed comparison of AUTC values for the studied *Aspergillus* strains in response to media and temperature is shown in Figure 2. Overall comparisons of the maximum OTA concentrations together with the day when the peak was achieved are in Appendix A.

OTA production was strongly influenced by both the cultivation medium and temperature. Among the tested media, YES proved to be the most suitable for the majority of *Aspergillus* strains, except for *A. carbonarius*, which produced the highest OTA levels on CYA. On YES, *A. ochraceus* strains were particularly productive, with strain no. 2 reaching 848.34 µg g^−1^ and strain no. 1 producing 92.36 µg g^−1^, both at 18 °C. *A. sulphureus* (no. 7) also reached its maximum on YES, producing 545.86 µg g^−1^ on day 21 at 18 °C. High OTA levels were also recorded for *A. westerdijkiae*, with strain no. 8 producing 479.74 µg g^−1^ on day 10 at 18 °C, and strain no. 9 reaching 591.28 µg g^−1^ on day 10 at 25 °C. *A. albertensis* (no. 3) yielded 197.45 µg g^−1^ on YES on day 21 at 30 °C. In contrast, *A. carbonarius* strains (nos. 4, 5, 6) were weak OTA producers overall, with the highest value being 19.90 µg g^−1^ for strain 4 on day 6 at 18 °C. On PDA, the highest OTA concentration was again recorded for *A. sulphureus* (7), with 551.94 µg g^−1^ on day 21 at 25 °C and 518.93 µg g^−1^ at 22˚C. *A. westerdijkiae* strains also showed strong production on PDA, with strain 8 reaching 311.58 µg g^−1^ and strain 9 producing 137.26 µg g^−1^ on day 10 at 25 °C. *A. albertensis* (no. 3) achieved 70.04 µg g^−1^ on day 6 at 30 °C, while *A. ochraceus* strains produced only low levels (5.66 and 38.65 µg g^−1^). *A. carbonarius* strains remained poor producers on PDA, with concentrations ranging from <LOD to 6.29 µg g^−1^. On CYA, *A. sulphureus* (no. 7) produced 246.79 µg g^−1^ on day 21 at 22˚C. This medium was most favorable for *A. carbonarius*, while *A. westerdijkiae* strains produced notably lower OTA levels than on PDA (88.00 and 79.83 µg g^−1^). MEA was the least suitable medium, with the maximum values not exceeding 63.37 µg g^−1^ for *A. sulphureus* (no. 7) and 45.06 µg g^−1^ for *A. abertensis* (no. 3).

Temperature also had a clear effect on OTA production. The highest concentrations were observed at 18 °C and 25 °C, while the lowest values were recorded at 30 °C. This temperature dependence was further evident in the individual isolates: *A. ochraceus* strains nos. 1 and 2 reached their maximum on YES at 18 °C, *A. albertensis* (no. 3) on YES at 30 °C, *A. sulphureus* (no. 7) on YES at 18 °C, *A. carbonarius* (no. 4) on YES at 18 °C, and *A. westerdijkiae* strains nos. 8 and 9 on YES, with strain no. 8 peaking at 18 °C and strain no. 9 at 25 °C.

### 3.4. Temporal Variability in OTA Concentration

During the experiment we found significant variations in the timepoints when maximum concentrations were achieved. The observed differences in the timepoints of maximum OTA production reflect distinct metabolic dynamics among the studied strains. Certain isolates, such as *A. ochraceus* (no. 2) and *A. albertensis* (no. 3), exhibited a continuous increase in OTA levels throughout the cultivation period. In these cases, toxin accumulation was progressive, since the rate of biosynthesis exceeded degradation or utilization, resulting in particularly high concentrations measured on day 30. In contrast, other strains reached a peak concentration at an earlier stage (e.g., on days 6, 10, or 21), after which OTA levels declined.

This pattern suggests that OTA is not always a stable end product during the entire cultivation cycle. The decrease in concentration may be explained by several processes, e.g., enzymatic degradation or transformation of OTA by the fungus itself, chemical or physical instability of OTA in the medium, influenced by factors such as pH, temperature, or oxidative reactions, or metabolic adaptation of the strain, where OTA biosynthesis is downregulated once growth enters a stationary phase and other metabolic pathways become prioritized.

These findings indicate that the ability to maintain stable OTA production is not universal, but rather strain- and species-dependent. While some isolates may be considered persistent accumulators, capable of producing OTA continuously over extended periods, others act as pulse producers, synthesizing the toxin rapidly to a maximum level before its subsequent decline. This variability highlights the importance of considering both the kinetics of production and potential degradation pathways when assessing the overall risk of OTA contamination.

## 4. Discussion

The aim of the study was to determine differences in the dynamics of OTA production by different isolates of the genus *Aspergillus* in response to various media and cultivation temperature. Fungal growth and OTA production are known to be strongly influenced by abiotic factors, particularly temperature and water availability, which affect germination, growth, and secondary metabolism. In our study, the highest OTA concentrations were generally observed at 18 °C and 25 °C, whereas production decreased markedly at 30 °C. This agrees with the tendency described by Esteban et al. [26], who found higher OTA levels at lower incubation temperatures, and with earlier work of Abarca et al. [27] and Alborch et al. [28]. It is worth noting, however, that temperature optima are not identical across studies, which may reflect subtle methodological differences or the natural variability of fungal isolates, as also shown by Medina et al. [29].

Considerable variability was observed not only between species but also among isolates of the same species. For example, *A. ochraceus* (no. 2) produced much higher OTA amounts than isolate no. 1, highlighting the well-documented intraspecific variability described by Varga et al. [16] and Freire et al. [17]. Interestingly, our *A. carbonarius* strains were among the weakest producers, despite being reported as major contributors to OTA contamination in grapes and wine elsewhere [30,31]. This discrepancy likely reflects the pronounced strain-dependent variability recently emphasised in studies of black aspergilli [17,32]. Several authors have documented that even genetically similar *A. carbonarius* isolates may differ dramatically in their ochratoxigenic potential due to differences in gene cluster regulation, metabolic downregulation under suboptimal conditions, or strain-specific detoxification pathways [30]. In addition, methodological factors may contribute to the observed differences, including the composition of the culture media, incubation duration, extraction efficiency, and analytical sensitivity, all of which have been shown to influence measured OTA yields [33]. All these differences may result either from genetic variations (presence or absence of toxin biosynthesis genes, gene clusters and regulators) or from differences in cultivation conditions, particularly temperature, duration, and growth medium [33,34]. Finally, the strictly in vitro character of our experiment may itself have contributed to lower OTA biosynthesis in *A. carbonarius*, as ecological stimuli present in natural substrates—such as microbial competition or nutrient stress—have been reported to enhance secondary metabolism in some black aspergilli [35]. It should also be noted that species identification in this study was carried out using macroscopic and micromorphological characteristics according to established taxonomic keys. While this approach is widely used for food-associated *Aspergillus* isolates, molecular identification would provide higher taxonomic resolution, especially within species complexes of sections *Circumdati* and *Nigri*. The absence of molecular confirmation represents a methodological limitation that should be addressed in future work to refine species delimitation and further support strain-level comparisons.

Culture media also had a strong influence on OTA output. Many authors have studied the production of OTA on various culture media [1,16,36,37,38] etc. Most *Aspergillus* strains produced the highest concentrations on YES medium, except for *A. carbonarius*, which favored CYA. Esteban et al. [26] reported similar results for *A. carbonarius*, while Ciegler and Fennell [39], Skrinjar and Dimić [40], and Lund and Frisvad [41], and others identified YES as optimal for a wide range of *Aspergillus* and *Penicillium* isolates. However, Filtenborg et al. [42] emphasized that the addition of trace metals and magnesium sulfate is required to support maximal OTA synthesis in this medium. Our observations therefore fit into the broader consensus, although the variability in nutrient requirements underlines that there is no universal “best” medium. Similarly, Wang et al. [43] reported that specific substrate components play a crucial role in modulating OTA production, further emphasizing the importance of nutrient composition in shaping ochratoxigenic potential. These findings are also consistent with recent studies demonstrating strong effects of carbon and nitrogen sources and a_w_ on OTA biosynthesis [44]. It seems more accurate to conclude that the suitability of a medium depends both on species identity and on the particular isolate. Although some isolates and media (particularly MEA) yielded low OTA concentrations, these combinations were intentionally retained because they represent natural variability within food-associated *Aspergillus* populations. Including weakly toxigenic isolates is essential for identifying both high- and low-risk strains and avoiding bias toward artificially strong OTA producers.

Species-specific trends became clear in our dataset. *A. sulphureus* (no. 7) proved to be the strongest OTA producer, confirming its potential risk profile. The two *A. westerdijkiae* strains (nos. 8 and 9) isolated from Slovak grape berries also showed high ochratoxigenic potential, consistent with reports linking this species to OTA contamination in coffee and meat products [15,45]. In contrast, *A. albertensis* produced only moderate amounts of OTA, in agreement with Bayman et al. [36]. These results remind us that species reputation alone is insufficient for risk assessment—individual isolates may diverge considerably from the expected pattern.

Another useful perspective came from evaluating the area under the toxin accumulation curve (AUTC). This parameter integrates toxin output over the entire cultivation period rather than focusing on a single peak value. Notably, the strains with the highest maximum OTA concentrations—*A. sulphureus*, *A. ochraceus* no. 2, and the *A. westerdijkiae* strains—also maintained production over time, yielding the largest AUTC values. This consistency across methods gives us greater confidence in their classification as high-risk producers. Still, the predictive value of AUTC should be interpreted with some caution, as it may underestimate sporadic but very high peaks of production. A major novelty of this study is the standardized comparison of nine *Aspergillus* isolates cultivated under identical controlled conditions, combined with the use of the AUTC parameter, which provides a more comprehensive view of long-term OTA dynamics than single-time-point measurements typically reported in previous studies.

Taken together, our findings highlight the complex interplay between isolate-level variability, culture conditions, and environmental factors in determining OTA biosynthesis. The pronounced variability observed even under controlled laboratory in vitro conditions reinforces the need to test multiple isolates and to interpret OTA production within the broader physiological context of each strain. From a practical perspective, these insights have immediate value for routine laboratory practice, particularly in the selection of stable and high-yielding fungal isolates for the production of OTA reference standards. Reliable reference material production requires predictable toxin output, and our results identify strains and conditions that support consistent OTA biosynthesis. Furthermore, incorporating both maximum concentration and cumulative production parameters (AUTC) may improve risk assessment strategies, especially for commodities such as grapes, coffee, and cereals that are vulnerable to OTA contamination [46].

Limitations: A key limitation of this study is the use of a single biological replicate for each condition. Although three technical replicates were performed and effect sizes were large, biological replication would further strengthen the reliability of the findings. Future research should therefore aim to validate high-risk strains using multiple biological replicates and broader environmental scenarios. Another methodological limitation is the absence of molecular identification for most isolates. Although phenotypic and micromorphological criteria remain widely used in food mycology, DNA-based approaches would provide higher taxonomic resolution and should be incorporated in future studies. However, additional molecular analyses cannot be performed retrospectively because several of the original isolates are no longer available in our laboratory collection; thus, only the morphological dataset obtained at the time of experimentation can be reported. Additionally, the experiments were conducted exclusively under controlled laboratory conditions using pure fungal cultures. This strictly in vitro approach does not account for biological interactions present in natural food matrices, such as competition with other microorganisms or substrate-specific stressors, which may significantly influence fungal growth and OTA production. Consequently, the applicability of the results to real-world food systems should be interpreted with caution, and future studies should incorporate more complex, naturally occurring substrates.

## 5. Conclusions

Based on the obtained results, the OTA-producing abilities of the *Aspergillus* strains varied considerably across all studied factors. OTA production was strongly strain-dependent, with notable variability even between isolates of the same species. Culture medium and temperature had significant effects, with most strains producing the highest OTA levels on YES medium at 18 °C or 25 °C, while *A. carbonarius* reached its maximum on CYA. Strains also differed in the timing of maximal production, reflecting distinct growth dynamics.

By evaluating both maximum OTA concentrations and cumulative production (AUTC), the strongest ochratoxigenic isolates were identified as *A. ochraceus* no. 2, *A. sulphureus* no. 7, and the two *A. westerdijkiae* strains (nos. 8 and 9) isolated from Slovak grapes. This combined approach provides a more comprehensive assessment of OTA biosynthesis and enables more accurate identification of high-risk strains. These findings are relevant for food safety risk assessment and for selecting stable, high-yielding isolates for OTA reference material production.

To further improve the prediction of OTA behaviour beyond laboratory conditions, future studies should investigate the molecular mechanisms underlying OTA biosynthesis and degradation and incorporate more complex, naturally occurring substrates.

## Figures and Tables

**Figure 1 microorganisms-13-02850-f001:**
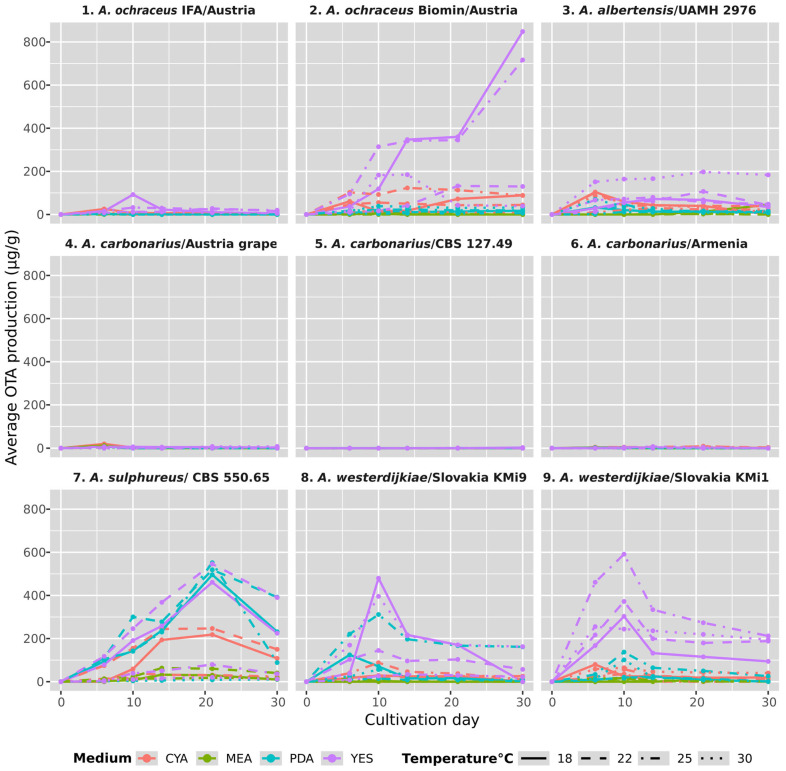
Overall comparison of maximum OTA production by *Aspergillus* strains across all tested cultivation media and conditions. Notes: *A.—Aspergillus*, MEA—malt extract agar, CYA—Czapek yeast agar, PDA—potato dextrose agar, YES—yeast extract sucrose agar.

**Figure 2 microorganisms-13-02850-f002:**
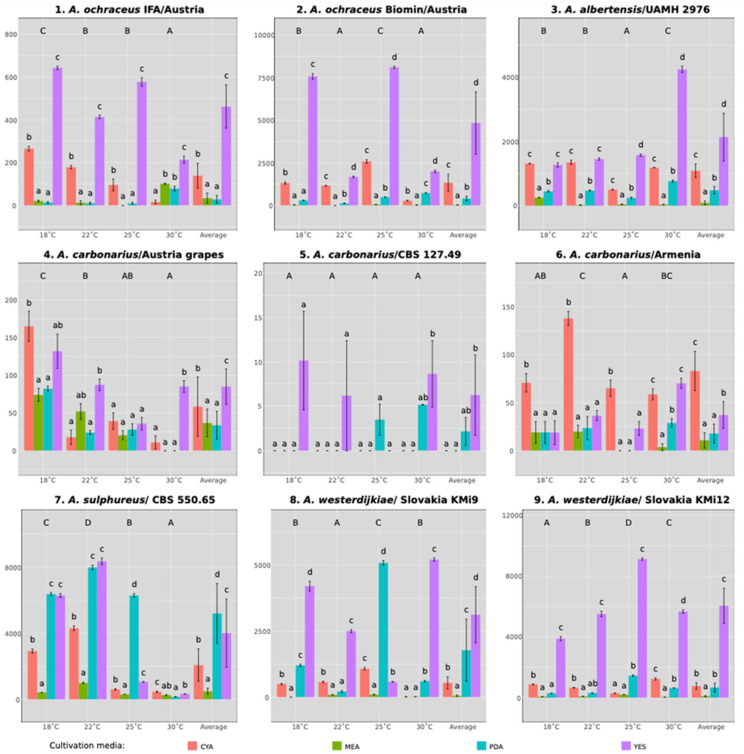
Comparison of AUTC values of OTA production by *Aspergillus* strains. Notes: *A.—Aspergillus*, CYA—Czapek yeast agar, MEA—malt extract agar, PDA—potato dextrose agar, YES—yeast extract sucrose agar. Letters above columns (within each temperature) indicate the difference between averages according to Tukey HSD at a significance level of 0.05. Uppercase letters indicate Tukey HSD differences between temperatures.

**Table 1 microorganisms-13-02850-t001:** An overview of *Aspergillus* strains.

Strain No.	Identification/Section	Source/Origin	Strain No.	Identification/Section	Source/Origin
1	*A. ochraceus */Circumdati*	cereals/IFA/Austria	6	*A. carbonarius */Nigri*	grape berries/Armenia
2	*A. ochraceus */Circumdati*	cereals/Biomin/Austria	7	*A. sulphureus* ^3^*/Circumdati* CBS 550.68	soil/India
3	*A. albertensis* ^1^*/Flavi* UAMH 2976	ear swab/Canada	8	*A. westerdijkiae* ^4^*/Circumdati* KMi9	grape berries/Slovakia
4	*A. carbonarius */Nigri*	grape berries/Austria	9	*A. westerdijkiae* ^5^*/Circumdati* KMi12	grape berries/Slovakia
5	*A. carbonarius* ^2^*/Nigri* CBS 127.49	coffee seeds/unknow			

Notes: *A.*—*Aspergillus*, CBS—Centraalbureau voor Schimmelcultures, Utrecht, Netherlands; IFA—Department für Agrarbiotechnologie, BOKU, Tulln, Austria; KMi—Collection of Department of Microbiology/Institute of Biotechnology, Slovak University of Agriculture in Nitra; UAMH—Microfungus Collection and Herbarium, University of Alberta, Canada; * wild strains of the genus *Aspergillus*, ^1^ cross reference: ATCC 58745, IMI 300485, NRPL 20602, GenBank: AY32066.1; ^2^ cross reference: IMI 313489, NRRL 4871, WB 4871, GenBank: MH868003.1; ^3^ cross reference: ATCC 16893, IMI 211397, LCP 89.2593, WB 4077, GenBank: OL772726; ^4^ GenBank:PX682725; ^5^ GenBank: PX682726.

**Table 2 microorganisms-13-02850-t002:** Morphological characteristics of wild *Aspergillus* strains use in this study.

Strain No.	Identification	Conidiophore/ Ornamentation of Wall *	Conidial Color *	Conidial Shape/ Ornamentation *	Sclerotia/ Color *	Growth at 37 ± 1 °C **	Growth on CREA/Acid Production
1	*A. ochraceus*	biseriate/rough	ochre	globose/finally rough	+/white	yes	poor/no
2	*A. ochraceus*	biseriate/rough	ochre	globose/finally rough	+/white	yes	poor/no
4	*A. carbonarius*	biseriate/smooth	black	globose/rough	-	yes	poor/strong
6	*A. carbonarius*	biseriate/smooth	black	globose/rough	-	yes	poor/strong

Notes: * on CYA (Czapek yeast extract agar) at 25 °C, ** on CYA, CREA—creatine sucrose agar, -: sclerotia absent, +: sclerotia present.

**Table 3 microorganisms-13-02850-t003:** Parameters of OTA production and accumulation of *Aspergillus* strains: Area under trapezoidal curve (AUTC).

Strain No.	Strain	Medium	t (°C)	*n*	AUTC	SD	Tukey HSD
1	*A. ochraceus* IFA/Austria	YES	25	3	577.21	34.75	b
2	*A. ochraceus* Biomin/Austria	YES	25	3	8113.56	113.62	e
3	*A. albertensis* UAMH 2976	YES	30	3	4253.07	169.12	c
4	*A. carbonarius* Austria, grapes	CYA	18	3	165.02	34.56	ba
5	*A. carbonarius* CBS 127.49	YES	18	3	10.14	9.62	a
6	*A. carbonarius* Armenia	CYA	22	3	137.75	12.92	a
7	*A. sulphureus* CBS 550.65	YES	22	3	8348.9	357.84	e
8	*A. westerdijkiae* Slovakia/KMi9	YES	30	3	5219.62	109.76	d
9	*A. westerdijkiae* Slovakia/KMi12	YES	25	3	9123.15	131.3	f

Notes: *A.—Aspergillus*, CYA—Czapek yeast agar, YES—yeast extract sucrose agar, *n*—replicates, t—cultivation temperature, SD—Standard deviation, AUTC—area under the trapezoidal curve, maximum production or AUTC values followed the same letter (in columns Tukey HSD) are not statistically significantly different at 0.05 significance level according to Tukey test.

**Table 4 microorganisms-13-02850-t004:** Parameters of OTA production and accumulation of *Aspergillus* strains: Average maximum production, day of maximum production.

Strain No.	Strain	Medium	t (°C)	Max. Production OTA (μg g^−1^)	SD	Tukey HSD	Max. Day
1	*A. ochraceus* IFA/Austria	YES	18	92.36	06.60	b	10
2	*A. ochraceus* Biomin/Austria	YES	18	848.34	10.32	f	30
3	*A. albertensis* UAMH 2976	YES	30	197.45	29.43	c	21
4	*A. carbonarius* Austria, grapes	CYA	18	19.90	02.23	a	6
5	*A. carbonarius* CBS 127.49	YES	18	02.90	02.75	a	30
6	*A. carbonarius* Armenia	CYA	25	09.28	02.10	a	21
7	*A. sulphureus* CBS 550.65	PDA	25	551.94	21.07	e	21
8	*A. westerdijkiae* Slovakia/KMi9	YES	18	479.74	32.12	d	10
9	*A. westerdijkiae* Slovakia/KMi12	YES	25	591.28	05.97	e	10

Notes: *A.—Aspergillus*, CYA—Czapek yeast agar, PDA—potato dextrose agar, YES—yeast extract sucrose agar, *n*—replicates, t—cultivation temperature, SD—Standard deviation, maximum production values followed the same letter (in columns Tukey HSD) are not statistically significantly different at 0.05 significance level according to Tukey test.

## Data Availability

The raw data supporting the conclusions of this article will be made available by the authors on request.

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
