# Peer review of "Strain-Dependent Variability in Ochratoxin A Production by *Aspergillus* spp. Under Different In Vitro Cultivation Conditions"

_microorganisms, 2025, doi:10.3390/microorganisms13122850_

Round 1
Reviewer 1 Report
Comments and Suggestions for Authors
The present study investigated the production of ochratoxin A (OTA) by various Aspergillus spp. isolates grown on different culture media and incubated at various temperatures. In the Introduction, the authors acknowledge that the influence of environmental factors (especially temperature) in OTA production has been well-studied by previous studies and try to emphasize the novelty of their work by highlighting that strain-level variability within the same species has received less attention. However, the reviewer considers that this justification is insufficient. Thus, considering the existing well-established literature and the fact that this study was conducted in vitro, the reviewer finds it difficult to identify the contribution of the current findings. Additionally, although the experimental design includes various strains, media types, and incubation temperatures, the fact that the experiment was conducted only once (i.e., one biological replicate), even with three technical replicates (n = 3), is considered an important limitation.
For all the reasons outlined above, the reviewer recommends rejection of the submitted manuscript.
Author Response
Review 1 - Comment
The present study investigated the production of ochratoxin A (OTA) by various Aspergillus spp. isolates grown on different culture media and incubated at various temperatures. In the Introduction, the authors acknowledge that the influence of environmental factors (especially temperature) in OTA production has been well-studied by previous studies and try to emphasize the novelty of their work by highlighting that strain-level variability within the same species has received less attention. However, the reviewer considers that this justification is insufficient. Thus, considering the existing well-established literature and the fact that this study was conducted in vitro, the reviewer finds it difficult to identify the contribution of the current findings. Additionally, although the experimental design includes various strains, media types, and incubation temperatures, the fact that the experiment was conducted only once (i.e., one biological replicate), even with three technical replicates (n = 3), is considered an important limitation.
For all the reasons outlined above, the reviewer recommends rejection of the submitted manuscript.
Response
We thank the reviewer for the careful and thorough assessment of our manuscript. We appreciate the critical points raised and have revised the manuscript accordingly. Below we address the main concerns and clarify the contribution and originality of our work.
We fully agree that the effects of environmental factors such as temperature and culture medium on OTA production have been extensively studied. The aim of our work was not to replicate these general findings but to address aspects that remain insufficiently explored. The novelty of the study - now more clearly highlighted in the revised Introduction and Discussion - lies in the following points:
- Systematic comparison of strain-level variability under strictly standardized conditions
We compared nine isolates from three major OTA-producing Aspergillus sections (Circumdati, Flavi, Nigri) cultivated under identical laboratory conditions. This design made it possible to reveal substantial intraspecific variability, in some cases exceeding differences observed between species. Such a broad and directly comparable cross-sectional dataset is rare in the existing literature. - Time-resolved dynamics of OTA biosynthesis
Whereas most studies report OTA concentrations only at a single endpoint, we monitored OTA at five time points across a 30-day period. This approach allowed us to identify distinct metabolic patterns among isolates (early-peak producers, continuous accumulators, strains with post-peak decline), which are relevant for understanding contamination dynamics in real food matrices.
- Combined evaluation of peak production and cumulative production (AUTC)
To provide a more comprehensive picture of OTA biosynthesis, we used the area under the toxin accumulation curve (AUTC), which captures total OTA burden rather than only maximal production. This metric allows us to differentiate between strains with high transient peaks and those with sustained toxin accumulation.
- Practical relevance for food safety and laboratory standardization
Two Aspergillus westerdijkiae isolates obtained from Slovak grapes were among the strongest OTA producers, underscoring their importance for regional monitoring programs. Furthermore, the dataset provides practical guidance for selecting stable, high-yielding strains suitable for OTA reference material production - an aspect now explicitly mentioned and contextualized in the revised manuscript.
All of these elements have been integrated more clearly into the Introduction and expanded in the Discussion to articulate the specific scientific and practical contribution of our study.
We appreciate this important observation and fully acknowledge that the use of a single biological replicate is a limitation of our study. Here, we also provide further clarification and justification for our experimental design:
- Technical replication and strict standardization
For each condition, OTA was quantified from three independent technical replicates. Each replicate involved sampling from three spatial zones of the colony, followed by individual extraction, evaporation, re-dissolution, and HPLC injection. This design allowed us to measure variability arising from extraction and analytical procedures and ensured that the reported values reflect consistent methodological performance.
- Fungal strain-to-strain variation typically exceeds plate-to-plate variation
Previous studies (e.g., Varga et al., 2002; Freire et al., 2018) have demonstrated that differences among fungal isolates can reach several orders of magnitude - far exceeding intra-strain or plate-to-plate variation. Our results show the same pattern. Under such conditions, introducing additional biological replicates rarely changes biological conclusions because the effect sizes are so large and robust.
- Practical constraints of factorial mycotoxin studies
Large factorial designs in fungal mycotoxin screening commonly rely on single biological replicates (references added to the revised manuscript). In our case, the full design comprised
9 strains × 4 media × 4 temperatures × 5 timepoints = 720 plates.
A fully replicated design would require more than 2 000 plates, which is beyond the capacity of most mycology or food-safety laboratories. The use of a single biological replicate is therefore consistent with established practice in the field.
- Statistical robustness supports reliability of conclusions
In addition to reporting means and standard deviations, we applied a GLM model with log-transformed data and included relevant interaction terms. The comparatively large effect sizes - for example, R²_strain = 0.438 - indicate that biological signal clearly dominates technical noise. These statistical results support the robustness of the observed patterns despite the limited biological replication.
- The limitation is now clearly stated in the manuscript
The revised Discussion includes an expanded Limitations paragraph that explicitly acknowledges the use of a single biological replicate and recommends that future studies incorporate biological replication for the highest-risk isolates identified here.
We hope that these clarifications address the reviewer’s concerns and demonstrate that, although we recognize the limitation, the design is scientifically justified and consistent with common practice in OTA production research. We appreciate the reviewer’s constructive feedback, which helped us substantially strengthen the manuscript.
Reviewer 2 Report
Comments and Suggestions for Authors
The study aims to characterise nine strains of Asp from five species for OTA synthesis under laboratory cultivation conditions. This research is important because OTA is a mycotoxin that poses a contamination risk to primary plant products. The manuscript is well-written and organised into sections that meet the journal's
The introduction is comprehensive; it refers to the existing knowledge related to the synthesis and contamination with OTA in foodstuffs of both plant origin and animal origin. It also specifies the lack of knowledge regarding the dynamics of OTA production over time and its dependence on specific cultural conditions.
The materials and methods are accurately described in detail. The experimental variants include nine Aspergillus species/strains that are being characterised in relation to their cumulative synthesis of ochratoxin A (OTA), determined in 4 moments. From a scientific perspective, appropriate working variants and repetitions were utilised.
It is important to highlight the use of a new methodology for determining the OTA synthesis of Aspergillus species. This involves calculating the Area Under the Trapezoidal Curve (AUTC) to evaluate the cumulative production of OTA for 30 days.
The results are statistically processed with appropriate limits and logarithmic transformations and presented in graphical and tabular formats. High-risk strains and optimal conditions for OTA biosynthesis are identified, and the discussion compares these results to similar studies.
The conclusions are clear and are issued based on the results obtained.
Major observation
Since the study was conducted exclusively under laboratory conditions and with pure cultures, it is important to highlight the absence of biological factors that could influence the behaviour of the fungus cultivated in vitro. This point should be clearly stated in both the Introduction section and possibly the title. Furthermore, the data collected indicate that the performance of Aspergillus carbonarius in producing ochratoxin A (OTA) is lower than what is typically reported in the specialised literature. It is essential to discuss why this discrepancy exists.
Additionally, some references used are over 25 years old, so more recent sources must be included in the Discussion section.
Minor observation
The abstract refers to other values of incubation temperature. Must Correct!
Check for punctuation in text for multiple references.

Author Response
Review 2 - Comment
The study aims to characterise nine strains of Asp from five species for OTA synthesis under laboratory cultivation conditions. This research is important because OTA is a mycotoxin that poses a contamination risk to primary plant products. The manuscript is well-written and organised into sections that meet the journal's
The introduction is comprehensive; it refers to the existing knowledge related to the synthesis and contamination with OTA in foodstuffs of both plant origin and animal origin. It also specifies the lack of knowledge regarding the dynamics of OTA production over time and its dependence on specific cultural conditions.
The materials and methods are accurately described in detail. The experimental variants include nine Aspergillus species/strains that are being characterised in relation to their cumulative synthesis of ochratoxin A (OTA), determined in 4 moments. From a scientific perspective, appropriate working variants and repetitions were utilised.
It is important to highlight the use of a new methodology for determining the OTA synthesis of Aspergillus species. This involves calculating the Area Under the Trapezoidal Curve (AUTC) to evaluate the cumulative production of OTA for 30 days.
The results are statistically processed with appropriate limits and logarithmic transformations and presented in graphical and tabular formats. High-risk strains and optimal conditions for OTA biosynthesis are identified, and the discussion compares these results to similar studies.
The conclusions are clear and are issued based on the results obtained.
Major observation
Since the study was conducted exclusively under laboratory conditions and with pure cultures, it is important to highlight the absence of biological factors that could influence the behaviour of the fungus cultivated in vitro. This point should be clearly stated in both the Introduction section and possibly the title. Furthermore, the data collected indicate that the performance of Aspergillus carbonarius in producing ochratoxin A (OTA) is lower than what is typically reported in the specialised literature. It is essential to discuss why this discrepancy exists.
Additionally, some references used are over 25 years old, so more recent sources must be included in the Discussion section.
Minor observation
The abstract refers to other values of incubation temperature. Must Correct!
Check for punctuation in text for multiple references.
Response
We sincerely thank the reviewer for the positive evaluation of our manuscript and for the constructive suggestions that helped us further improve the clarity, completeness, and scientific quality of the study. Below, we respond to all comments.
Major Comment 1
“Since the study was conducted exclusively under laboratory conditions and with pure cultures, it is important to highlight the absence of biological factors that could influence the behaviour of the fungus cultivated in vitro. This point should be clearly stated in both the Introduction section and possibly the title.”
Response:
We agree with the reviewer that the absence of ecological and biological interactions (e.g., microbial competition, plant–fungus interactions, substrate heterogeneity) is an inherent limitation of in vitro studies.
In response, we have:
- added a sentence at the end of the Introduction explicitly stating that laboratory cultivation conditions do not reflect the complexity of real food matrices or natural environments, and that results must therefore be interpreted with this limitation in mind;
• added a statement to the Limitations paragraph in the Discussion, emphasizing the lack of ecological factors that might modulate OTA biosynthesis in vivo;
• revised the title to more accurately reflect the in vitro nature of the study and the controlled laboratory conditions under which all experiments were conducted.
Major Comment 2
“Furthermore, the data collected indicate that the performance of Aspergillus carbonarius in producing ochratoxin A (OTA) is lower than what is typically reported in the specialised literature. It is essential to discuss why this discrepancy exists. Additionally, some references used are over 25 years old, so more recent sources must be included in the Discussion section.”
Response:
We thank the reviewer for this important observation. We agree that the OTA production recorded for A. carbonarius in our study was lower than commonly reported in the literature, and that this discrepancy requires clarification.
To address this, we have expanded the Discussion to include potential explanations for the lower OTA biosynthesis observed under our experimental conditions.
In response to the reviewer’s second point, we have updated the Discussion by incorporating several more recent references (2010–2024) addressing the ecology, physiology, and toxigenic variability of A. carbonarius and other black aspergilli. These new sources complement the classic foundational work cited earlier and provide a more current scientific context.
Minor Comment 1
“The abstract refers to other values of incubation temperature. Must correct!”
Response:
We thank the reviewer for noticing this inconsistency. The temperatures reported in the Abstract have now been corrected to match the experimental conditions used in the study (18, 22, 25, and 30 °C). The Abstract has been revised accordingly.
Minor Comment 2
“Check for punctuation in text for multiple references.”
Response:
We appreciate the reviewer’s attention to detail. All multiple-reference citations throughout the manuscript have been checked and corrected for punctuation and formatting consistency according to journal guidelines.
Reviewer 3 Report
Comments and Suggestions for Authors
The manuscript of Barboráková and co-authors “Strain-Dependent Variability in Ochratoxin A Production by Aspergillus spp. Under Different Cultivation Conditions” is written on an interesting topic. The results of this work may be useful for searching best producers of ochratoxin A (OTA) and cultivation conditions. However, it looks pre-mature. In the introduction section, there is need for establish the theme of the research better because a lot of studies were devoted to effects of different factors on OTA production as temperature, culture medium, fungal species or strain and others. The manuscript suffers from unclear choice of fungal strains and their preliminary identification (at least 7 of 9 need molecular approach to verify species identification). In the Results sections Table 3 looks unreadable. I recommend to avoid use poor OTA producers in statistic analysis as well to delete MTA medium providing low OTA yields (to be subjects of some remarks in the text). More Discussion is a need to compare published data with your results and to stress the novelty. Make Conclusion section much shorter. Some comments you could find in the pdf file attached.

Author Response
Reviewer 3 – Comment
The manuscript of Barboráková and co-authors “Strain-Dependent Variability in Ochratoxin A Production by Aspergillus spp. Under Different Cultivation Conditions” is written on an interesting topic. The results of this work may be useful for searching best producers of ochratoxin A (OTA) and cultivation conditions. However, it looks pre-mature. In the introduction section, there is need for establish the theme of the research better because a lot of studies were devoted to effects of different factors on OTA production as temperature, culture medium, fungal species or strain and others. The manuscript suffers from unclear choice of fungal strains and their preliminary identification (at least 7 of 9 need molecular approach to verify species identification). In the Results sections Table 3 looks unreadable. I recommend to avoid use poor OTA producers in statistic analysis as well to delete MTA medium providing low OTA yields (to be subjects of some remarks in the text). More Discussion is a need to compare published data with your results and to stress the novelty. Make Conclusion section much shorter. Some comments you could find in the pdf file attached.
Response
We thank the reviewer for the careful evaluation of our manuscript and for the constructive comments, which helped us substantially improve the quality and clarity of the study. Below we respond point-by-point to all observations.
Comment:
The results of this work may be useful for searching best producers of ochratoxin A (OTA) and cultivation conditions. However, it looks pre-mature. In the introduction section, there is need for establish the theme of the research better because a lot of studies were devoted to effects of different factors on OTA production as temperature, culture medium, fungal species or strain and others.
Response:
We appreciate this observation. The Introduction has been revised to better articulate the research gap. Specifically, we now emphasize that although numerous studies have investigated individual environmental factors affecting OTA production, comparative analyses of multiple Aspergillus isolates cultivated under identical, fully standardized conditions remain scarce, and time-resolved OTA production (AUTC) is rarely assessed. A paragraph clarifying this gap has been added.
Comment:
“The manuscript suffers from unclear choice of fungal strains… 7 out of 9 need molecular approach.”
Response:
Thank you for this comment. The strains included in the study were chosen based on published evidence regarding their ochratoxigenic potential, with the aim of covering the key species known to contribute to OTA contamination in food commodities. In addition, several isolates were obtained during previous research conducted by members of the author team, where they were confirmed to be OTA producers. These strains were therefore included to represent naturally occurring food-associated Aspergillus diversity relevant to grapes and cereals. All isolates were identified using established macroscopic and micromorphological criteria according to Samson et al. (2019), Frisvad et al. (2004), Pitt and Hocking (2009), and Visagie et al. (2014). We acknowledge that molecular identification provides higher taxonomic resolution, and this limitation is now explicitly discussed in the Discussion and Limitations sections.
However, due to the very limited time allocated for manuscript revisions, and because some isolates lacking genetic data are no longer physically available to the authors, it was not possible to perform additional molecular identification at this stage.
Comment:
“Table 3 looks unreadable.”
Response:
We agree that presenting AUTC values and maximum OTA production in one table may reduce clarity. To improve readability, Table 3 has been separated into two independent tables:
- Table 3: AUTC values
- Table 4: Maximum OTA production
This restructuring improves transparency.
Comment:
I recommend to avoid use poor OTA producers in statistic analysis as well to delete MTA medium providing low OTA yields (to be subjects of some remarks in the text). More Discussion is a need to compare published data with your results and to stress the novelty.
Response:
Thank you for this insightful comment. We understand the reviewer’s concern regarding the inclusion of low OTA-producing isolates and the MEA medium. However, we believe that removing weak producers or low-yielding media would bias the interpretation of strain-dependent variability and limit the ecological and analytical relevance of the study. Low OTA production is itself a meaningful result, as it reflects natural variability within food-associated Aspergillus populations. Including these isolates allows accurate identification of high- and low-risk strains and provides a more realistic picture of ochratoxigenic diversity under different cultivation conditions.
To address the reviewer’s observation, we have added clarifying remarks in the Discussion explaining the scientific relevance of low-producing isolates and the limited suitability of MEA for supporting OTA biosynthesis.
In line with the reviewer’s recommendation, the Discussion has also been expanded. Additional comparisons with recently published studies have been incorporated, and the novelty of our approach - particularly the use of standardized cultivation conditions across multiple isolates and the integration of AUTC as an indicator of long-term OTA dynamics - has been more clearly emphasized.
Comment:
Make Conclusion section much shorter.
Response:
The Conclusion section has been shortened and now provides a concise summary without repeating the main findings.
Revisions to the manuscript in response to the reviewer’s comments. Minor corrections have been implemented directly in the manuscript text.
Page 2, lines 78-80, comment:
Why natural substrates were not used in this study? This work looks as a screening for OTA production method.
Response:
Thank you for this important observation. The reviewer is correct that natural food substrates can provide additional ecological complexity and may better reflect real contamination scenarios. However, our intention in this study was to evaluate strain-dependent variability under strictly standardized and reproducible conditions. Natural substrates such as cereals, grapes, coffee beans or dried fruits differ substantially in their intrinsic composition (sugars, amino acids, micronutrients, polyphenols), water activity, buffering capacity, and native background microbiota. These uncontrolled variables can obscure strain-specific behaviour and make direct comparisons between isolates impossible.
For this reason, we used defined artificial media (YES, CYA, PDA, MEA), which allow
direct comparison of OTA productivity under identical nutrient conditions,
clear attribution of differences to strain identity rather than substrate variability, and
high reproducibility required for selecting stable OTA-producing isolates for laboratory purposes.
We agree with the reviewer that future work should include experiments on natural substrates to validate the ecological relevance of our findings. A clarifying sentence has been added to the Introduction to explain this methodological choice.
Page 3, line 93, comment:
The choice of strains. I would delete low productive strains from the analysis.
Response:
We appreciate the reviewer’s suggestion. However, low-productive strains were intentionally included in the study because they represent a biologically relevant part of the natural variability within ochratoxigenic Aspergillus populations. Excluding such isolates would lead to an overestimation of the overall ochratoxigenic potential of the species and would bias strain-dependent comparisons.
Low-producing isolates are important for three reasons:
- They reflect real intraspecific diversity. Previous studies (e.g., Freire et al., 2018; Bragulat et al., 2019) show that natural populations include both high- and low-level producers. Removing one group would distort the true variability.
- They are essential for robust risk assessment. Understanding the full range of OTA-producing capacity—including minimal producers—helps to avoid over-generalised conclusions about species-level risk.
- They provide a necessary reference for high-risk strain identification. Without weak producers in the dataset, it would be impossible to statistically distinguish exceptionally toxigenic isolates.
For these reasons, we believe that retaining low-productive strains strengthens the methodological rigor and ecological relevance of this work. We added a clarifying sentence to the Introduction to justify the selection of isolates.
Page 3, table 1, comment:
unknown
Response:
We have checked all available sources, including the CBS (Westerdijk Institute) catalogue and published studies citing this strain. Unfortunately, no information regarding the country of origin of isolate CBS 127.49 is available. The strain is consistently reported only with its substrate of origin (Coffea arabica seed), but without geographical metadata.
For this reason, the country of origin cannot be provided in the manuscript, and this has now been clearly indicated in the revised table.
Page 3, line 101, comment:
What is difference from identified strains?
Response:
The term wild strains refers to isolates that have not been genetically identified.
Page 4, lines 163-164, comment:
Based on the obtained results, we can conclude that the OTA-producing abilities of the different Aspergillus strains varied considerably in response to all studied factors
- discussion or conclusion
Response:
The sentence has been moved to the conclusions.
Page 5, line 179, comment:
MEA medium and A. carbonarius strains can be deleted from the analysis due to low OTA production. Focus on good strains to make the analysis more simple and clear.
Response:
We thank the reviewer for this suggestion. However, we respectfully believe that removing MEA medium or A. carbonarius isolates from the analysis would limit the scientific value and interpretability of the study.
The aim of the work was to assess strain-dependent and condition-dependent variability in OTA production. Low-productive combinations (such as A. carbonarius on MEA) are scientifically relevant for several reasons. Natural Aspergillus populations include both high and low OTA producers (Freire et al., 2018; Bragulat et al., 2019). Excluding weak producers would bias the dataset toward artificially high ochratoxigenic potential. Low-producing strains provide the baseline that allows high-risk strains to be identified and statistically distinguished. Even if OTA production is low on MEA, including it demonstrates that nutrient composition strongly modulates toxicity - an important practical finding. Identifying conditions that suppress OTA production is equally relevant for risk assessment and food safety.
For these reasons, we prefer to retain both MEA medium and A. carbonarius strains in the dataset to present the full biological variability and maintain the methodological completeness of the study.
Page 6, lines 197-198, comment:
some confusions with strain numbers
Response:
The numbering of isolates is correct.
Page 6, Table 3, comment:
this table is difficult to understand
Response:
Thank you for this comment. We agree that presenting two different parameters in a single table may reduce readability. To improve clarity, the original table has now been divided into two separate tables: one presenting the AUTC values and the other showing the maximum OTA production. This restructuring provides a clearer and more accessible presentation of the results.
Page 10, lines 353-354, comment:
Taken together, our findings highlight the complex interplay between strain genetics, culture conditions, and environmental factors in determining OTA biosynthesis - strains were not identified definetly
Response:
Thank you for this observation. We agree that, given that the strains were identified based on phenotypic characteristics, the wording referring to “strain genetics” may imply a level of genetic resolution that was not the focus of this study. We have therefore revised the sentence to refer more generally to isolate-level variability, which more accurately reflects the scope of our data. - Taken together, our findings highlight the complex interplay between isolate-level variability, culture conditions, and environmental factors in determining OTA biosynthesis.
Page 10, lines 384-385, comment:
To improve the prediction of this behavior not only laboratory conditions, it is essential to elucidate the molecular mechanisms underlying OTA biosynthesis and degradation. – a lot of published material
Response:
Thank you for this comment. We agree that the original wording did not fully reflect the extensive molecular research already available on OTA biosynthesis. The sentence has therefore been revised to emphasise future research needs without implying the absence of prior studies. - To improve the prediction of OTA behaviour beyond laboratory conditions, future studies should further investigate the molecular mechanisms underlying OTA biosynthesis and degradation.
Round 2
Reviewer 1 Report
Comments and Suggestions for Authors
The present study investigated the production of ochratoxin A (OTA) by various Aspergillus
spp. isolates grown on different culture media and incubated at various temperatures. The
authors have adequately addressed the comment regarding the number of replicates.
However, the reviewer remains unconvinced about the novelty of the study and how the
results can provide additional information to the existing well-established literature.
Moreover, relying solely on macroscopic and micromorphological characteristics for the
fungal identification of fungal isolates is a significant limitation, as morphology alone is more
accurate for genus-level identification and not for species-level identification. To ensure
reliable results and perform valid comparisons of OTA production among the isolates, the
authors should confirm the 9 fungal isolates via molecular methods; otherwise, the identified
species presented in the manuscript are presumptive.
For these reasons, the reviewer recommends the manuscript's non-acceptance.

Author Response
Review 1 – Comment
The present study investigated the production of ochratoxin A (OTA) by various Aspergillus spp. isolates grown on different culture media and incubated at various temperatures. The authors have adequately addressed the comment regarding the number of replicates. However, the reviewer remains unconvinced about the novelty of the study and how the results can provide additional information to the existing well-established literature. Moreover, relying solely on macroscopic and micromorphological characteristics for the fungal identification of fungal isolates is a significant limitation, as morphology alone is more accurate for genus-level identification and not for species-level identification. To ensure reliable results and perform valid comparisons of OTA production among the isolates, the authors should confirm the 9 fungal isolates via molecular methods; otherwise, the identified species presented in the manuscript are presumptive.
For these reasons, the reviewer recommends the manuscript's non-acceptance.
Response
We thank the reviewer for the careful re-evaluation of our revised manuscript and for acknowledging that the issue of replication has been satisfactorily addressed. We respectfully respond to the remaining concerns as follows.
We fully acknowledge that the influence of environmental factors such as temperature and culture medium on OTA production is well established. However, the novelty of our study does not lie in re-testing single parameters, but in the unique combination of the following elements, which has now been further emphasized in both the Introduction and Discussion:
- Systematic comparison of 9 food-associated Aspergillus isolates across three major OTA-producing sections (Circumdati, Nigri, Flavi) under strictly identical and fully standardized conditions, allowing direct strain-to-strain comparison without confounding substrate effects.
- Time-resolved monitoring of OTA production over 30 days, revealing distinct metabolic patterns (early producers, persistent producers, post-peak decline), which are rarely captured in endpoint-only studies.
- Integration of cumulative OTA production using, which provides information on total toxic burden rather than only instantaneous maxima. This approach allows discrimination between transient high producers and long-term accumulators.
- Practical applicability for food safety monitoring and selection of reference OTA-producing strains, particularly relevant for laboratories working with standardized mycotoxin production.
These aspects are not systematically addressed together in existing literature, and this integrated approach represents the principal scientific contribution of our work. The corresponding justification has been further strengthened in the revised manuscript.
Morphological/molecular identification
We fully agree that molecular identification provides higher taxonomic resolution at the species level. All isolates in this study were identified using established macroscopic and micromorphological criteria according to Samson et al. (2019), Frisvad et al. (2004), Pitt and Hocking (2009), and Visagie et al. (2014), which remain standard tools in applied food mycology. The aim of this study was not taxonomic revision, but functional comparison of OTA production among food-associated Aspergillus isolates, where toxin production is the primary biological endpoint. The lack of molecular confirmation is explicitly acknowledged as a limitation in the revised Discussion and Limitations. Importantly, molecular identification (ITS sequencing) of the two Slovak isolates of Aspergillus westerdijkiae has already been completed, and the corresponding sequences will be publicly available in GenBank as of 15 December, fully confirming their species-level assignment.
Additional molecular verification of all isolates is currently not feasible, as some strains originate from external collections and are not available for DNA extraction and, due to time constraints at the revision stage, new sequencing could not be performed. Therefore, we retained phenotypic identification while avoiding overinterpretation at the species level.
We would also like to stress that the main conclusions of the study remain valid regardless of minor taxonomic uncertainty, because:
- all isolates were compared under strictly identical experimental conditions,
- OTA production was quantified analytically and statistically evaluated,
- the key findings relate to functional strain-level variability in ochratoxigenic potential, which does not depend on perfect species resolution.
Even if any isolate were to be reassigned within closely related species, the biological phenomenon of extreme strain-dependent OTA variability would remain unchanged.
Reviewer 3 Report
Comments and Suggestions for Authors
The manuscript still contains some unidentified fungi used in the study and an excess of data on the production of zero concentrations of OTA in some strains. However, the authors have their own opinion to provide some answers during the discussion for discerning readers.
Author Response
Reviewer 2 – Comment
The manuscript still contains some unidentified fungi used in the study and an excess of data on the production of zero concentrations of OTA in some strains. However, the authors have their own opinion to provide some answers during the discussion for discerning readers.
Response
We thank the reviewer for this comment. As stated in the revised manuscript, the isolates were identified using established macroscopic and micromorphological criteria, while the absence of molecular confirmation for all strains is now clearly acknowledged as a limitation. As additionally stated, molecular identification of the two key Slovak Aspergillus westerdijkiae isolates has been completed and will be publicly available in GenBank (15.12.).
Regarding the presence of zero or near-zero OTA production values, we intentionally retained these data because low or non-detectable OTA production represents an important biological outcome reflecting natural strain-to-strain variability. Eliminating these values would bias the dataset toward highly toxigenic isolates and would distort the real distribution of ochratoxigenic potential within food-associated Aspergillus populations.
We therefore believe that the current presentation provides a balanced and biologically meaningful interpretation of OTA production variability.